# Design of a Switchable Filter for Reflectionless-Bandpass-to-Reflectionless-Bandstop Responses

**DOI:** 10.3390/mi14020424

**Published:** 2023-02-10

**Authors:** Gangxiong Wu, Hao Wu, Wei Qin, Jin Shi, Wei Zhang, Longlong Lin, Qian Li

**Affiliations:** 1School of Information Science and Technology, Nantong University, Nantong 226019, China; 2Research Center for Intelligent Information Technology, Nantong University, Nantong 226019, China; 3Nantong Key Laboratory of Advanced Microwave Technology, Nantong University, Nantong 226019, China; 4Zhongtian Radio Frequency Cable Co., Ltd., Nantong 226010, China; 5School of Communications and Information Engineering & Artificial Intelligence, Xi’an University of Posts and Telecommunications, Xi’an 710121, China

**Keywords:** reflectionless responses, switchable filter, microstrip line, bandpass mode, bandstop mode

## Abstract

In this paper, a switchable filter based on the microstrip line (ML) with reflectionless-bandpass-to-reflectionless-bandstop responses is designed, theoretically validated, and fabricated. This single-port reflectionless bandpass filter (R-BPF) consists of a BPF and a shunt-connected bandstop section with terminated absorption resistors. The single-port reflectionless bandstop filter (R-BSF) is made of a BSF and a parallel bandpass circuit with terminated absorption resistors. These two reflectionless operational modes, namely R-BPF and R-BSF, are allowed to reconfigure the multifunctional filtering device using PIN diodes. In addition, a theoretical analysis of terminal impedance is performed to illustrate the working mechanism of the reflectionless response. To demonstrate the application of the proposed designs, a prototype of the switchable filter for R-BPF to R-BSF responses is fabricated and measured. For the R-BPF mode, the 3-dB fractional bandwidth (FBW) is 36.75% (1.67–2.42 GHz) with a 10-dB reflectionless bandwidth (RBW) of 1.36–2.58 GHz (i.e., FBW of 61.9%). For the R-BSF mode, the 10-dB bandwidth is 13% (1.85–2.11 GHz) with a 10.7-dB RBW of 1–3 GHz (i.e., FBW of 100%). An acceptable agreement between the measured and simulated results has been achieved.

## 1. Introduction

With the rapid development of modern 5G/6G wireless communication systems, high-performance requirements with increasingly compact reconfigurable microwave passive devices are in tremendous demand. Switchable filters have been widely discussed in recent years because they can meet the requirements of RF terminal circuits in multiple operating frequencies, operating modes, and complex electromagnetic environments [1,2,3]. Based on this, a series of switchable filters with different operating modes has been reported in the literature [4,5,6,7,8,9,10,11,12,13,14,15,16]. These designs mainly use different types of RF switches to change the ON and OFF states of circuits, namely PIN diodes, microelectromechanical switches (MEMS), field effect transistors (FETS), amongst others. In addition, the *n*-pole *n*-throw (*n*P*n*T) switches allow the RF signals to be routed along different paths, such as single-pole double-throw (SPDT) switches and double-pole double-throw (DPDT) switches. A comparative summary of the performance of typical RF switches is given in Table 1. A frequency-agile bandstop-to-bandpass switchable filter based on a MEMS diode has been reported in [4]. The switchable filter is composed of tunable substrate-integrated cavity resonators, and bandstop-to-bandpass can be switched due to the inter-resonator coupling structures. In [5], a frequency-tunable tri-function filter has been designed using substrate-integrated waveguide (SIW) resonators, which can exhibit three different frequency responses: all-pass, bandstop, and bandpass. In addition, different kinds of multifunctional switchable filters using a microstrip-lines (MLs) structure have been presented in [6,7,8,9]. All these switchable modes are actually realized by turning the PIN switches ON/OFF to dynamically connect or disconnect transmission lines and stubs so that multiple modes can be selected discretely. Although these designs realize multiple filtering modes, these schemes reflect out-of-band signals to the source, which generally causes inevitable interference [17]. Based on this fact, absorptive circuits have been widely used in various microwave passive devices, such as the reflectionless bandpass filter (R-BPF) [18,19], reflectionless filtering power divider (R-FPD) [20], reflectionless bandstop filter (R-BSF) [21,22], reflectionless low-pass filter (R-LPF) [23], amongst others. An all-band R-BPF based on MLs and series-resistor-connected stubs was demonstrated in [19]. The absorptive circuits not only eliminate out-of-band reflected signals but also play an important role in achieving the passband flat group delay (GD) response. In [21], a high stopband-rejection and good frequency-selectivity R-BSF is proposed. A coupled line (CL) with an open-circuited stub and CLs loaded at the ports are used to obtain a broad stopband response. A single grounded resistor is utilized to absorb the undesired signals in the stopband.

In recent years, absorptive circuits have been integrated into the switchable filter to avoid the impact of reflected signals in the stopband region on the stable operation of the circuit [24,25,26]. Specifically, through a fixed absorptive circuit, the reflectionless response of one of the different modes can be achieved. For instance, a switchable filter for the bandpass-to-absorptive-bandstop response has been reported in [24]. Two single-pole double-throw (SPDT) switches are used to control the operating modes. The absorptive bandstop response is achieved by tuning the Q-factor of SIW resonators. In [26], a switchable R-BSF to BPF using a dual-mode ring resonator has been reported. A PIN diode was used to connect and disconnect the additional perturbing stub and to achieve an absorptive bandstop response and mode conversion. However, these designs cannot achieve a reflectionless response for all modes and can only achieve the reflectionless response of one of them, because the additional absorption circuit is difficult to apply to different modes. Therefore, as in conventional reflective-type filters, the non-transmitted input signal in their bandstop region cannot be dissipated inside themselves and reflected to the source.

In this paper, a switchable filter based on MLs with R-BPF-to-R-BSF modes is proposed. The common part of the BPF, the BSF, and the absorptive circuit are fully used to realize the fusion of the R-BPF and R-BSF functional filters. By switching RF PIN diodes ON/OFF, this proposed topology could provide R-BPF and R-BSF modes. Meanwhile, a detailed theoretical analysis is presented, respectively, to explain the working mechanism. Finally, a prototype of the switchable filter for R-BPF-to-R-BSF responses is fabricated and measured, and an acceptable agreement between the measured and simulated results is achieved. Overall, the proposed design achieves two switchable functionalities: reflectionless bandpass and reflectionless bandstop responses. To the best of the authors’ knowledge, few published works on switchable filters can realize both R-BPF and R-BSF modes. The reflectionless switchable filter proposed in this paper can not only realize the reflectionless response of all operating modes but also has the advantages of a wide bandwidth, wider absorption bandwidth, and lower insertion loss. The proposed design in this paper has the potential to be applied in RF front-end nonlinear chains, such as power amplifiers and mixers, to improve the behavior of adjacent active stages of their associated transceivers and strengthen the expansibility of communication systems.

## 2. Analysis of the Switchable R-BPF-to-R-BSF

The original idea of the proposed design comes from the fusion of the R-BPF and R-BSF with similar topologies and from the consolidation of common stubs to simplify the scheme. The non-reflection filter circuits are based on the duplexer topology and consist of two-channel filters with complementary transfer functions. The main channel is connected to the input and output terminals, one end of the auxiliary channel is connected to the input terminal, and the other end is connected to the grounding resistance. The configuration of the proposed switchable R-BPF to R-BSF, which consists of a switchable filtering circuit and a switchable absorptive circuit, is shown in Figure 1. The switchable filtering circuit incorporates two transmission paths with a bandpass (Path 1)/bandstop (Path 2) response. Two pairs of PIN diodes named *D*_1_/*D*_1′_ and *D*_2_/*D*_2′_ are employed for the switching of the two filtering modes. Furthermore, *Z_c_*_1_ is connected before the BPF and BSF circuits, and its role is to adjust the matching of filtering circuits and absorptive circuits. The proposed switchable absorptive circuit functions through the inclusion of a lossy open-ended stub at the input connection point of the in-parallel transmission-lines (TLs) arrangement to realize the reflectionless property. The TLs (*Z_c_*_2_–*Z_c_*_4_) and resistances (*R*_1_ and *R*_2_) are common parts of switchable absorptive circuits, which means that these parts will be used no matter which absorptive mode works. Here, two pairs of PIN diodes named *S*_1_/*S*_1′_ and *S*_2_ are used to switch absorptive modes. Except for the fact that TLs (*Z_p_*_3_, *Z_p_*_6_, *Z_s_*_5_, and *Z_p_*_7_) are half-wavelength (2*θ* = 180°), the electrical length of all the other TL segments is *θ* = 90° at the center frequency *f*_0_.

### 2.1. R-BPF Mode

When the PIN switches *D*_1_/*D*_1′_, *S*_1_/*S*_1′_ are turned to the ON state and the others remain in the OFF state, the proposed scheme is reconfigured to the R-BPF mode, as shown in Figure 2. The BPF section is mainly established by two cascaded open-ended stubs (*Z_p_*_1_–*Z_p_*_6_) connected in parallel at the terminal of a half-wavelength TL (*Z_p_*_7_). The absorptive circuit section is established by the common parts and in-parallel TL segments (*Z_r_*_1_–*Z_r_*_3_) at the terminal of the TL (*Z_c_*_4_). The *Z_in_*_1_ and *Z_in_*_2_ marked in Figure 2 represent the input impedance of the BPF circuit and absorptive circuit, respectively.

In order to satisfy the reflectionless characteristics of the single-port R-BPF, the input impedance (*Z_in_*_1_ and *Z_in_*_2_) of the BPF and absorptive circuit section must meet the following condition:(1)$$Z_{in1}//Z_{in2}\to Z_{0}$$
where *Z*_0_ = 50 Ω is the port impedance.

By analyzing the *Z_in_*_1_ in Figure 2, the transfer ABCD matrix of the BPF section can be obtained in Equation (2) as:(2)$$M_{BPF}=\left[\begin{array}{cc} A_{1} & B_{1}\\ C_{1} & D_{1} \end{array}\right]=\left[\begin{array}{cc} \cos\theta & jZ_{c1}\sin\theta \\ j\frac{1}{Z_{c1}}\sin\theta & \cos\theta \end{array}\right]\left[\begin{array}{cc} 1 & 0\\ \frac{1}{Z_{stub1}} & 1 \end{array}\right]\left[\begin{array}{cc} \cos2\theta & jZ_{p7}\sin2\theta \\ j\frac{1}{Z_{p7}}\sin2\theta & \cos2\theta \end{array}\right]\left[\begin{array}{cc} 1 & 0\\ \frac{1}{Z_{stub2}} & 1 \end{array}\right]$$
where:(3)$$Z_{stub1}=\frac{j[Z_{p1}Z_{p2}(Z_{p1}+Z_{p2})\sin2\theta -2Z_{p3}Z_{p1}(Z_{p2}\cos^{2}\theta -Z_{p1}\sin^{2}\theta )\cot2\theta ]}{\left(Z_{p1}+Z_{p2})Z_{p3}\cos2\theta +2Z_{p2}(Z_{p1}\cos^{2}\theta -Z_{p2}\sin^{2}\theta \right)}$$
(4)$$Z_{stub2}=\frac{j[Z_{p4}Z_{p5}(Z_{p4}+Z_{p5})\sin2\theta -2Z_{p6}Z_{p4}(Z_{p5}\cos^{2}\theta -Z_{p4}\sin^{2}\theta )\cot2\theta ]}{\left(Z_{p4}+Z_{p5})Z_{p6}\cos2\theta +2Z_{p5}(Z_{p4}\cos^{2}\theta -Z_{p5}\sin^{2}\theta \right)}$$

According to Formulas (2)–(4) and classical two-port network theory, the input impedances *Z_in_*_1_ of the BPF section can be obtained as:(5)$$Z_{in1}=\frac{[(\cos\theta +j\frac{Z_{c1}}{Z_{stub1}}\sin\theta )\cos2\theta -\frac{Z_{c1}}{Z_{p7}}\sin\theta \sin2\theta +\frac{1}{Z_{stub2}}X]Z_{0}+X}{[(j\frac{1}{Z_{c1}}\sin\theta +\frac{1}{Z_{stub1}}\cos\theta )\cos2\theta +j\frac{1}{Z_{p7}}\sin2\theta \cos\theta +\frac{1}{Z_{stub1}}X]Z_{0}+Y}$$
where:(6)$$\left\{ \begin{array}{l} X=j[Z_{p7}\sin2\theta (\cos\theta +j\frac{Z_{c1}}{Z_{stub1}}\sin\theta )+Z_{c1}\sin\theta \cos2\theta ]\\ Y=jZ_{p7}(j\frac{1}{Z_{c1}}\sin\theta +\frac{1}{Z_{stub1}}\cos\theta )\sin2\theta +\cos\theta \sin2\theta \end{array}\right.$$

Next, the mathematical formula of the input impedance *Z_in_*_2_ of the absorptive circuit section will be analyzed. For the convenience of calculation, the equivalent circuit of the absorptive circuit is shown in the bottom right area of Figure 2. In this case, the ring circuit network consisting of TLs (*Z_r_*_1_–*Z_r_*_3_, *Z_c_*_4_) is replaced by a matrix ${\left[ABCD\right]}_{3}$. First, the ABCD matrix and Y matrix of the series TLs (*Z_r_*_1_–*Z_r_*_3_) can be expressed, respectively, as:(7)$$M_{zr}=\left[\begin{array}{cc} A_{4} & B_{4}\\ C_{4} & D_{4} \end{array}\right]=\left[\begin{array}{cc} \cos\theta & jZ_{r1}\sin\theta \\ j\frac{1}{Z_{r1}}\sin\theta & \cos\theta \end{array}\right]\left[\begin{array}{cc} \cos\theta & jZ_{r2}\sin\theta \\ j\frac{1}{Z_{r2}}\sin\theta & \cos\theta \end{array}\right]\left[\begin{array}{cc} \cos\theta & jZ_{r3}\sin\theta \\ j\frac{1}{Z_{r3}}\sin\theta & \cos\theta \end{array}\right]$$
(8)$$M_{zr}=\left[\begin{array}{cc} \;\;\frac{D_{4}}{B_{4}} & \frac{B_{4}C_{4}-A_{4}D_{4}}{B_{4}}\\ -\frac{1}{B_{4}} & \frac{A_{4}}{B_{4}} \end{array}\right]$$


At the same time, the TL (*Z_c_*_4_) can be characterized by the following Y matrix:(9)$$Y_{zc4}=\left[\begin{array}{c} \;\;\;\;\;\frac{1}{jZ_{zc4}}\cot\theta \;\;\;\;\;-\frac{1}{jZ_{c4}\sin\theta }\\ -\frac{1}{jZ_{c4}\sin\theta }\;\;\;\;\;\;\;\;\;\;\;\frac{1}{jZ_{zc4}}\cot\theta \end{array}\right]$$

The ring circuit network consisting of TLs (*Z_r_*_1_–*Z_r_*_3_, *Z_c_*_4_) can be characterized by the following Y matrix and ABCD matrix:(10)$$Y=\left[\begin{array}{c} Y_{11}\;\;\;\;Y_{12}\\ Y_{21}\;\;\;\;Y_{22} \end{array}\right]=Y_{zr}+Y_{zc4}$$
(11)$$M_{zinp3}=\left[\begin{array}{c} A_{3}\;\;\;\;B_{3}\\ C_{3}\;\;\;\;D_{3} \end{array}\right]=\left[\begin{array}{c} \;\;\;\;\;\;-\frac{Y_{22}}{Y_{21}}\;\;\;\;\;\;\;\;\;\;\;-\frac{1}{Y_{21}}\\ -\frac{Y_{22}Y_{21}-Y_{12}Y_{21}}{Y_{21}}\;\;\;\;-\frac{Y_{11}}{Y_{21}} \end{array}\right]$$

Therefore, the input impedance *Z_inp_* of the ring circuit network consisting of TLs (*Z_r_*_1_–*Z_r_*_3_, *Z_c_*_4_) can be calculated using the following equations:(12)$$Z_{inp}=\frac{A_{3}R_{2}+B_{3}}{C_{3}R_{2}+D_{3}}$$

The input impedance *Z_in_*_2_ of the absorptive circuit can be obtained as:(13)$$Z_{in2}=R_{1}+Z_{inp2}=R_{1}+Z_{c2}\frac{\left(Z_{inp1}+jZ_{c2}\tan\theta \right)}{\left(Z_{c2}+jZ_{inp1}\tan\theta \right)}$$
where:(14)$$Z_{inp1}=Z_{c3}\frac{\left(Z_{inp}+jZ_{c3}\tan\theta \right)}{\left(Z_{c3}+jZ_{inp}\tan\theta \right)}$$

According to Formulas (2)–(14), the mechanism of the R-BPF response can be further explored. The variation of the input impedance responses of the BPF (*Z_in_*_1_) and absorptive circuit (*Z_in_*_2_) can be obtained. For the verified presentation, the comparisons between numerical calculation results and Advanced Design System (ADS) simulation responses of the input impedance are shown in Figure 3a, and they are in good agreement. The shaded region represents the bandpass region for the R-BPF mode near the center frequency *f*_0_ and a bandstop for the absorptive circuit. It can be seen that they exhibit opposite impedance-response characteristics. The value of *Z_in_*_2_ approaches 50 Ω, while *Z_in_*_1_ tends to be infinite, and vice versa. In detail, a few observations can be made from Figure 3a:(1)At *f*_0_ (*θ* = 90°), the value of *Z_in_*_2_ approaches 50 Ω, while *Z_in_*_1_ tends to be infinite. The signal energy is transmitted through the BPF, but the path to the absorption circuit section is interrupted.(2)At 0.75*f*_0_ and 1.25*f*_0_ (*θ* = 67.5° & 112.5°), the value of *Z_in_*_2_ approaches infinite, while *Z_in_*_1_ tends to 50 Ω. That is to say, the signal cannot pass through the BPF and is reflected to the absorptive circuit, and then dissipated by the loading resistors.(3)Three transmission poles (*TP*1–*TP*3) can be obtained to achieve a considerable bandwidth. Four transmission zeros (*TZ*1–*TZ*4) in the lower and upper bands can be generated to achieve a good out-of-band suppression.

Thus, the values of the TLs are reasonably chosen by the derived closed formulae, as follows: *Z_c_*_1_ = 50 Ω, *Z_c_*_2_ = 140 Ω, *Z_c_*_3_ = 120 Ω, *Z_c_*_4_ = 100 Ω, *Z_p_*_1_ = 90 Ω, *Z_p_*_2_ = 135 Ω, *Z_p_*_3_ = 125 Ω, *Z_p_*_4_ = 88 Ω, *Z_p_*_5_ = 80 Ω, *Z_p_*_6_ = 98 Ω, *Z_p_*_7_ = 150 Ω, *Z_r_*_1_ = 157 Ω, *Z_r_*_2_ = 120 Ω, *Z_r_*_3_ = 98 Ω, *R*_1_ = 44 Ω, *R*_2_ = 150 Ω. The S-parameters of the R-BPF with ideal TLs are shown in Figure 3b. We can see that this R-BPF can achieve a 10-dB input-reflectionless spectral range from 0.75*f*_0_ to 1.25*f*_0_. In addition, as predicted above, three transmission poles (*TP*1–*TP*3) and four transmission zeros (*TZ*1–*TZ*4) are obtained.

### 2.2. R-BSF Mode

When the PIN switches 2 and 2′ are turned to the ON state and the other switches are turned off, the reconfigurable filter works in an R-BSF mode. Figure 4 illustrates the ideal circuit; the BSF circuit section consists of quarter-wavelength T-shaped TLs (*Z_s_*_1_–*Z_s_*_3_) and in-parallel TL segments (*Z_s_*_4_–*Z_s_*_6_). The absorptive circuit section is composed of common parts and an open stub (*Z_r_*_4_) to the parallel TL (*Z_c_*_4_). The *Z_in_*_3_ and *Z_in_*_4_ marked in Figure 4 represent the input impedance of the BSF and absorptive circuit section, respectively. In this case, the Y matrix is used to simplify the analysis, and the equivalent circuit of the BSF circuit is shown in the bottom right area of Figure 4.

According to the equivalent circuit, the transfer matrix *M_zin_*_3_ of the BSF section can be expressed as:(15)$$M_{zin3}={\left[\begin{array}{cc} A & B\\ C & D \end{array}\right]}_{5}=\left[\begin{array}{cc} \cos\theta & jZ_{c1}\sin\theta \\ j\frac{1}{Z_{c1}}\sin\theta & \cos\theta \end{array}\right]\left[\begin{array}{cc} -\frac{Y_{22}{}^{\prime }}{Y_{21}{}^{\prime }} & -\frac{1}{Y_{21}{}^{\prime }}\\ -\frac{\Delta Y^{\prime }}{Y_{21}{}^{\prime }} & -\frac{Y_{11}{}^{\prime }}{Y_{21}{}^{\prime }} \end{array}\right]$$
where the $\Delta Y^{\prime }=Y_{11}{}^{\prime }Y_{22}{}^{\prime }-Y_{12}{}^{\prime }Y_{21}{}^{\prime }$; based on Equations (7)–(12), the input impedance *Z_in_*_3_ of the BSF circuit section can be obtained as:(16)$$Z_{in3}=\frac{jZ_{c1}{}^{2}(\Delta Y^{\prime }Z_{0}+Y_{11}{}^{\prime })\sin\theta +Z_{c1}(1+Y_{22}{}^{\prime })\cos\theta }{j(Y_{22}{}^{\prime }+1)\sin\theta +(\Delta Y^{\prime }Z_{0}+Y_{11}{}^{\prime })Z_{c1}\cos\theta }$$

On the other hand, the input impedance *Z_in_*_4_ of the absorptive circuit section can be expressed as:(17)$$Z_{in4}=R_{1}+Z_{c2}\frac{Z_{inc3}+jZ_{c2}\tan\theta }{Z_{c2}+jZ_{inc3}\tan\theta }$$
where:(18)$$Z_{inc3}=Z_{c3}\frac{Z_{incs}+jZ_{c3}\tan\theta }{Z_{c3}+jZ_{incs}\tan\theta }$$
(19)$$Z_{incs}=\frac{Z_{r4}Z_{c4}(R_{2}+jZ_{c4}\tan\theta )}{jR_{2}(Z_{c4}+Z_{r4})\tan\theta +(Z_{r4}-Z_{c4}\tan\theta )Z_{c4}}$$

Consequently, the input impedances of the BSF section (*Z_in_*_3_) and absorptive circuit section (*Z_in_*_4_) can be obtained through Equations (15)–(19), respectively, and the curves of *Z_in_*_3_ and *Z_in_*_4_ are plotted in Figure 5a. The shaded region presents a bandstop for the BSF and a bandpass for the absorptive circuit. It can be found that the value of *Z_in_*_4_ approaches 50 Ω at *f*_0_, while *Z_in_*_3_ tends to be infinite, and vice versa. In detail, the following conclusions can be obtained from Figure 5a.

(1)At *f*_0_ (*θ* = 90°), the value of *Z_in_*_4_ approaches 50 Ω, while *Z_in_*_3_ tends to be infinite. The path to the BSF is interrupted, and the reflected signal is transmitted to the absorptive circuit.(2)At the lower and upper passbands, the value of *Z_in_*_4_ approaches infinite, while *Z_in_*_3_ tends towards 50 Ω. Thus, the signal can pass through the BSF, and the absorptive circuit does not work.(3)Four transmission poles (*TP*1–*TP*4) and one transmission zero can be obtained to achieve a good filtering characteristic. The positions of transmission poles are: *f_TP_*_1_ = 0.5*f*_0_, *f_TP_*_2_ = 0.75*f*_0_, *f_TP_*_3_ = 1.25*f*_0_, *f_TP_*_1_ = 1.5*f*_0_, and one transmission zero is: *f_TZ_*_1_ = *f*_0_.

Finally, the values of the TLs are reasonably chosen by the derived closed formulae, as follows: *Z_s_*_1_ = 103 Ω, *Z_s_*_2_ = 155 Ω, *Z_s_*_3_ = 45 Ω, *Z_s_*_4_ = 48 Ω, *Z_s_*_5_ = 69 Ω, *Z_s_*_6_ = 48 Ω, *Z_r_*_1_ = 157 Ω, *Z_r_*_4_ = 45 Ω. The S-parameters of the R-BSF with ideal TLs are shown in Figure 5b. It can be seen that this R-BSF can achieve a 10-dB input-reflectionless spectral range from 0.5*f*_0_ to 1.5*f*_0_. In addition, as predicted above, four transmission poles and one transmission zero are obtained.

## 3. Design Procedure

For the proposed switchable filter for reflectionless-bandpass-to-reflectionless-bandstop responses, the design procedure is summarized as follows.

Step 1: Get the initial values of ML segments (*Z_c_*_1_, *Z_p_*_1_~*Z_p_*_7_) of the BPF according to ADS and obtain the characteristic impedance Zin1 through Formulas (2)–(6).

Step 2: Determine the initial values of ML segments (*Z_c_*_2_~*Z_c_*_4_, *Z_r_*_1_~*Z_p_*_3_) and the resistance (*R*_1_, *R*_2_) of the absorptive circuit according to Formulas (7)–(14) to ensure that *Z_in_*_2_ and *Z_in_*_1_ have opposite impedance-response characteristics.

Step 3: Optimize the parameters of the absorption circuit in ADS to ensure that *S*_11_ is less than −10 dB.

Step 4: Obtain the initial values of the R-BSF according to the above Steps 1–3.

Step 5: Integrate R-BPF and R-BSF and further optimize parameters.

Step 6: Build the EM models in a full-wave environment and choose an appropriate substrate.

Step 7: Optimize the whole structure, fabricate a prototype, and conduct testing.

## 4. Measured Results and Discussion

Through ideal circuit model analysis using the normalized operating frequency above, the R-BPF and R-BSF modes can be obtained and switched by changing the states of PIN switches. Then, the electrical lengths and characteristic impedances of the ideal circuit model can be converted to their corresponding physical dimensions operating at the center frequency of 2 GHz. A prototype of the switchable filter for reflectionless-bandpass-to-reflectionless-bandstop responses has been fabricated and measured with the vector network analyzer N5230C. The substrate selected for the proposed filter prototype is RO4003C with a relative dielectric constant of *ε_r_* = 3.38, a loss tangent of *tanδ* = 0.0027, and a thickness of *h* = 1.524 mm. The values of the capacitors and the resistors employed in the biasing networks of the RF PIN diodes are *C* (*C*_1_–*C*_7_) = 100 pF and resistors *R* (*R_a_*_1_–*R_a_*_4_, *R_b_*_1_–*R_b_*_5_) = 1 kΩ, respectively. RF PIN diodes SMP1345-079LF from Skyworks were used for electronic diodes. Figure 6a,b show their design model (i.e., layout) and photographs, respectively. The blue cables and red cables are the power feeding lines of the R-BPF mode and R-BSF mode, respectively. The whole circuit area including the PIN diode switch circuit is 11.2 cm × 9 cm (i.e., 0.74 *λ_g_* × 0.6 *λ_g_*, where λ_g_ is the effective wavelength at *f*_0_).

The measured and simulation results of the two reflectionless filtering modes are provided in Figure 7 for comparison purposes. For the R-BPF mode, the 3-dB fractional bandwidth (FBW) is 36.7% (1.67–2.42 GHz) with an in-band return loss higher than 11.7 dB, minimum in-band insertion loss of 1.73 dB, and 10-dB reflectionless bandwidth (RBW) of 1.36–2.58 GHz (i.e., FBW of 61.9%). The passband group delay τ_21_ is from 1.1 to 2.4 ns, and the flat group delay is revealed in the central region of the passband. For the R-BSF mode, the 10-dB bandwidth is 13% (1.85–2.11 GHz) with an in-band return loss that is better than 35 dB, minimum insertion loss of 0.8 dB, and 10.7-dB RBW of 1–3 GHz (i.e., FBW of 100%). The passband group delay τ_21_ is from 0.9 to 2.1 ns. It can be seen that the results of the measurement and simulation generally have an acceptable agreement. They also show some differences from the comparison of simulated and experimental results of the proposed switchable reflectionless filter, the differences arising due to the possible machining size error and welding loss.

As a comparison with state-of-the-art designs, Table 2 illustrates the performance of the proposed design and some other related works. It can be seen that, compared with the former multifunctional switchable filters [6,7,8], the proposed reflectionless switchable filter has the advantages of a switchable reflectionless response, which means that two reflectionless filtering modes, R-BPF and R-BSF, can be obtained. At the same time, compared with the switchable filters with absorptive circuits [24,26], in addition to realizing the reflectionless function of all operating modes, it also has a wider filtering bandwidth, wider absorption bandwidth, and lower insertion loss.

## 5. Conclusions

In this paper, a switchable filter with reflectionless-bandpass-to-reflectionless-bandstop responses is designed and theoretically analyzed. A prototype is fabricated, and the measured results show that this proposed design agrees well with the predicted counterparts. Compared with state-of-the-art designs, the proposed filter can realize a reflectionless response for all operating modes. Most importantly, the proposed switchable reflectionless filter featured a wider filtering bandwidth, wider absorption bandwidth, and lower insertion loss. In practice, only one DC power is required to switch two reflectionless modes, but the number of control switches needs to be further reduced to simplify assembly. Despite all this, these properties indicate that this article shows great potential in practice and is highly demanded in further multi-function wireless circuits and multimode communication systems.

## Figures and Tables

**Figure 1 micromachines-14-00424-f001:**
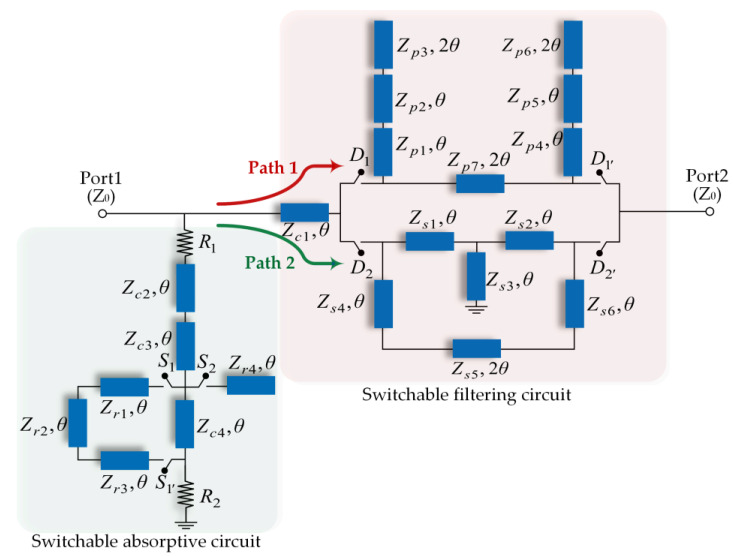
Configuration of the proposed switchable R-BPF-to-R-BSF.

**Figure 2 micromachines-14-00424-f002:**
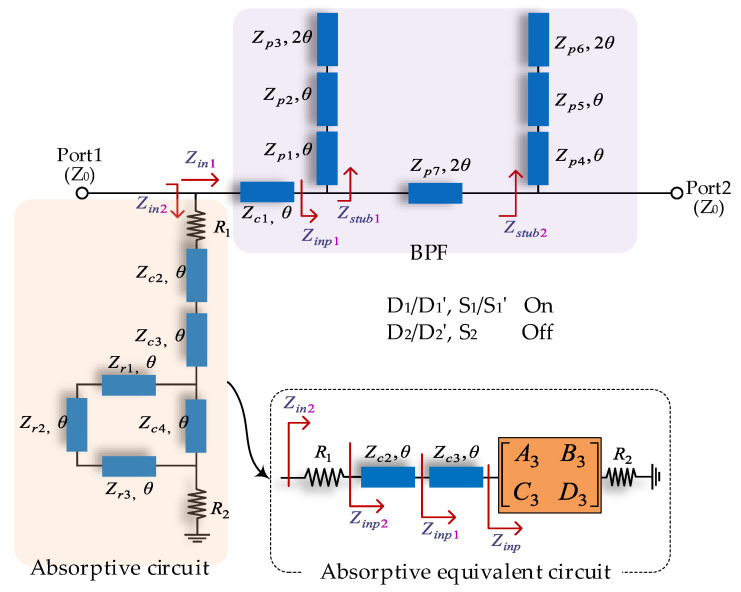
R-BPF mode of the switchable filter.

**Figure 3 micromachines-14-00424-f003:**
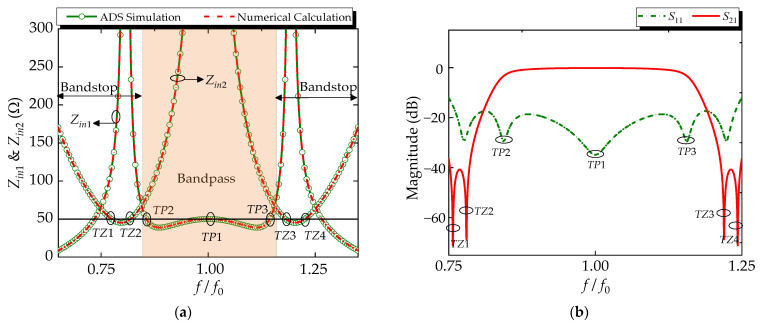
(**a**) The numerically calculated and simulated input impedance of absorptive structure (*Z_in_*_2_) and BPF (*Z_in_*_1_) of the proposed R-BPF; (**b**) S-parameters of the R-BPF with ideal TLs.

**Figure 4 micromachines-14-00424-f004:**
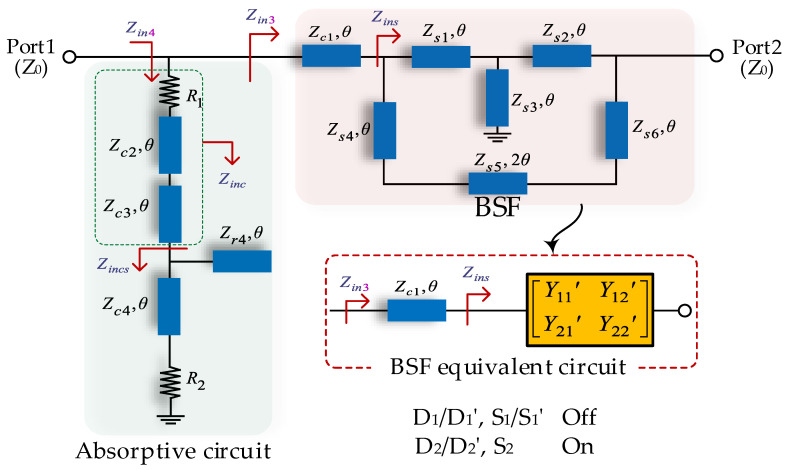
R-BSF mode of the switchable filter.

**Figure 5 micromachines-14-00424-f005:**
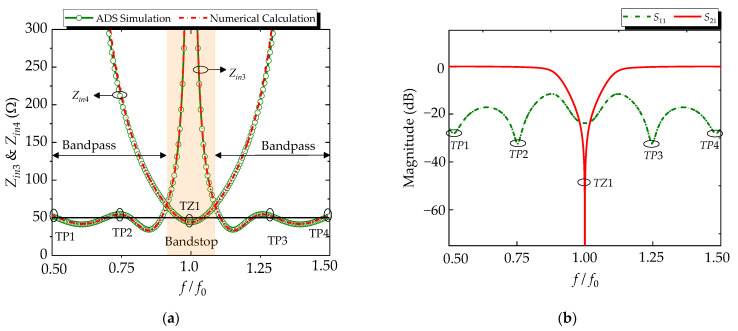
(**a**) The numerically calculated and simulated input impedance of absorptive circuit (*Z_in_*_4_) and BSF (*Z_in_*_3_) of the proposed R-BSF. (**b**) S-parameters of the R-BSF mode with ideal TLs.

**Figure 6 micromachines-14-00424-f006:**
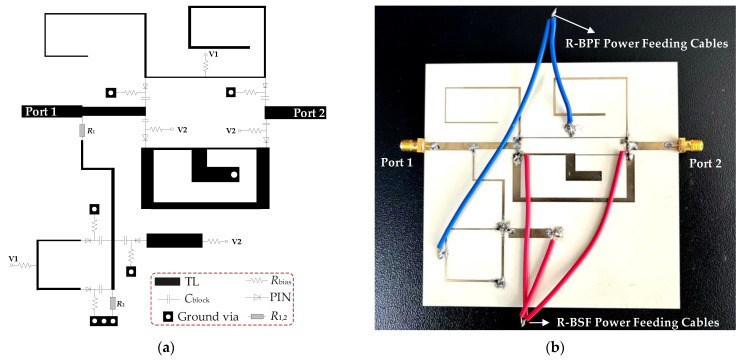
(**a**) Layout of the proposed switchable R-BPF-to-R-BSF; (**b**) fabricated photograph.

**Figure 7 micromachines-14-00424-f007:**
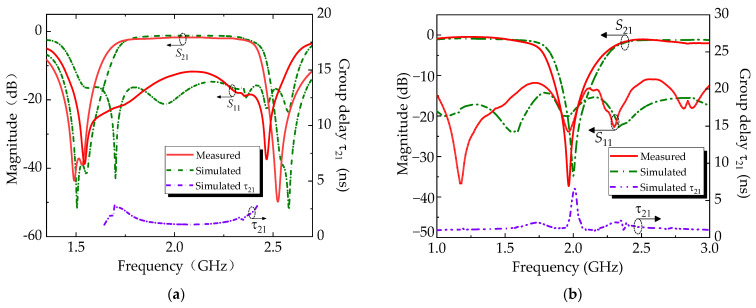
Measured and simulated results of the reflectionless switchable filter. (**a**) R-BPF mode; (**b**) R-BSF mode.

**Table 1 micromachines-14-00424-t001:** Typical performance data for RF switch.

Switch Type	Insertion Loss (dB)	Isolation (dB)	Voltage (V)	Current (mA)	Switching Time (ns)	Frequency (GHz)	Power Handling (W)
PIN diodes	0.3–1	>25	5–10	3–20	1–100	~40	~200
MEMS	0.4–2	>25	5–30	~0	1–20	~40	~10
MESFETs	0.05–2	>40	20–80	0	1000–40,000	~100	~1

(MESFETs: metal semiconductor FETs).

**Table 2 micromachines-14-00424-t002:** The proposed switchable reflectionless filter compared with other related works.

Ref.	Response	Reflectionless Modes	*f*_0_ (GHz)	FBW (%)	Insertion Loss (dB)	10-dB RBW (%)	Type of Switching
[6]	BPF/BSF/ DB-BPF	NO	1.92/1.92/1.58&2.23	53.1/30.2/3.8& 2.7	0.9/0.8/2&2.2	No	PIN
[7]	UWB-BPF/NB-BPF/BSF	NO	2.4/2.92/3	95.7/25.3/3	0.94/0.8/0.7	No	PIN
[8]	BPF/BSF/ DB-BSF/ASF	NO	1.92/2/1.61&2.36/2	9.7/50.2/21.1&20/ND	1.95/1.1/1.2&1.9/ND	No	PIN
[24]	BPF/R-BSF	One mode	2.16/2.58	2.95/<2	6.7/1.1	NO/30.1%	SPDT
[26]	BPF/R-BSF	One mode	2.4/2.4	26/<4	ND/ND	NO/75.5%	PIN
This work	R-BPF/R-BSF	All modes	2.05/1.98	36.7/13	1.73/0.8	61.9%/100%	PIN

DB-BPF: Dual-band BPF; UWB: Ultra-wideband; NB: narrow-band; ASF: all-stop filter; ND: not defined

## Data Availability

The data presented in this study are available on request from the first author.

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
