# Peer review of "Design of a Switchable Filter for Reflectionless-Bandpass-to-Reflectionless-Bandstop Responses"

_micromachines, 2023, doi:10.3390/mi14020424_

Round 1

Reviewer 1 Report

The authors proposed a switchable filter based on the microstrip line (ML) with reflectionless bandpass to reflectionless bandstop responses, and verify and fabricate it theoretically. The strength of this work lies in the simplicity of the proposed structure. However, this work lacks innovation, and this article needs further revision. I suggest a significant correction.

1. The corresponding indexes of the non-reflective switchable filter proposed by the author can all be satisfied in the relevant literature in the past, and the size designed is too large. In fact, in Table 1, it is difficult for me to distinguish the advantages of this structure compared with other structures from one of the parameters.

2. It is suggested that the author try to cite articles with high impact factors in the field in recent years.

3. It is suggested that the fonts of all the diagrams in the manuscript should be consistent with those in the text. The size of figure 5 (a) and 5 (b) should be the same.

4. It is suggested that the author accurately modify the format of the references. For example, in reference 2, Tsai, H.-J. In the format of Tsai, H.J. . In reference 11, the year 2015 needs to be bolded and so on.

5. Please explain the potential of the non-reflective switchable filter in the manuscript in practical applications.

Reviewer 2 Report

It is claimed that a switchable filter with reflectionless bandpass to reflectionless bandstop responses is theoretically designed. Also, measuring results are reported. The authors should address the following items.

1) Major concern: The authors have been directly proposed the filter configuration in Figure (1). The filter design process is not clear! In other words, what is the origin of the initial guess of the proposed filter configuration?

2) All frequency components of a signal are delayed when they pass through a filter. Group delay in a filter is the time delay of the signal through the device under test as a function of frequency. If we take the example of a modulated sine wave, for example an AM radio signal. Group delay is a measurement of the time taken by the modulated signal to get through the system. For an ideal filter, the phase will be linear and the group delay would be constant. However, in the real world group delay distortions occur, as signals at different frequencies take different amounts of time to pass through a filter. Could the authors report the phase of S11 and S21 or group delay?

3) It is needed a better description of the results. Please avoid giving very obvious explanations.

4) Please highlight the advantages and drawbacks of the proposed filter. Are there any drawbacks for the proposed filter?

5) Conclusion is intended to help the reader understand why your research should matter to them after they have finished reading the paper. A conclusion is not merely a summary of your points or a re-statement of your research problem but a synthesis of key points.

6) The authors should move Figure 1 toward page 3.

7) I couldn’t find R2 in Figure 1.

8) In lines 94 and 95, it is stated that all the TL segments have the electrical length \theta. However, in Figure 1, the length of Zp3, Zp7and Zs2 are 2*\theta!

9) Please check index of all components of Figure 1.

10) How the absorptive circuit is obtained? Couldn't the authors have used a simpler absorptive circuit?

11) How the numerical calculation results are obtained in Figure 3?

Reviewer 3 Report

Introduction – Please describe and explain more about the techniques used for switching. PIN Diode switches, SPDT switches, and a MEMS device were referenced. Please provide a comparison table of the various switching techniques including advantages and disadvantages. What is the reason for your selection of PIN diodes? What about MEMS switches or capacitive tuners, where the capacitive tuner could directly modulate the capacitors. What is the critical properties for the switch? How important is matching of the switch pairs?

What is the reason few published works have failed to achieve both R-BPF and R-BSF? Please expand on the references in this area. What is the reason for the success of the approach taken here? What is the primary advantage of this integration and area for application.

Analysis: How are the PIN diode switches modeled in the theoretical or ADS environment? What are the specs of the switches, especially the critical specifications. What is the impact of the switches on the overall system design?

How were the TLs modeled? How would this compare to field models?

What are the limiting assumptions in the models for both theoretical and ADS?

What are the spec targets for the filters and design targets for TLS and circuit elements. How were the values of the Capacitor and Resistor elements selected?

How important is matching of the elements especially stub1 and 2. Please explain more about the actual design of the switchable filter and the reason for the layout shown in Figure 6. Please explain more about the fabrication details of the substrate and TLs and SMA connection launches. Why was the RO4003C substrate selected? This affects the TL design, which should be explained in more detail.

Please provide more details about the differences between the model and measured results. Please explain how machining size error or welding loss explain this difference. Please provide a list and pareto of the sensitivity. Given the foundation of the numerical model and the ADS model. This sensitivity analysis can be easily accomplished. Please address uniform parameter variation and mismatch between elements such as Cs, Rs, TLs, and switches. Are there manufacturing or assembly challenges that would make this R-BPF/R-BSF prohibitive?

Good figures

Good plots of simulated vs numerical

Good plots of measured vs simulated

Sufficient details for equations

Table 1 is a good comparison of switchable filters

Reviewer 4 Report

The authors have designed a switchable filter with reflectionless bandpass and bandstop responses. The paper is well written and the obtained results have validated theory and simulation. However, the following modifications should be considered in the manuscript as follows:

-         Kindly indicate two transmission paths with the bandpass and bandstop responses in Figure 1.

-         More explanation is needed for selecting the values of transmission lines in both bandpass and bandstop conditions, as mentioned in lines 148-150, and also in lines 193-194.

-         Check if the D1, D2, S1, and S2 switches states are written correctly in figures 2 and 4.

-         The applications of the presented device in the designed frequency bands should be mentioned.

-         Read manuscript carefully and correct typos in the text and figures. For example, “dB” in Figure 7.

-         Add some explanations about group delay of the designed filter.

-         In table 1, the return loss “35 dB” is written mistakenly in the insertion loss column for the proposed and cited works. Kindly correct this mistake in the table. Both insertion loss and return loss for BPF and BSF modes should be clearly given in this table for all cited works.

-         How the main configuration of the proposed filter is obtained in Figure 1? Kindly explain the design procedures.

Round 2

Reviewer 1 Report

I don't think the article needs to be revised.

Reviewer 2 Report

Thanks the authors for answers and updates. The quality of the manuscript is extremely improved. However, I think that my first question is not answered correctly (In other words, the authors' answer is a bit vague for me). How the original idea of the proposed structure (Figure 1) is obtained? In the revised version, the authors stated only the steps of the design procedure. If it is possible, please provide more details or add suitable reference/references.

Reviewer 3 Report

Table 1 is a good comparison of switches. What are the most critical specifications of the switch for the filter? What are the tradeoffs? Do the PIN diodes limit performance

Reviewer 4 Report

All of my comments are addressed in the manuscript correctly by authors. The manuscript can now be accepted in the present form.
